# Regret Bounds for Online Portfolio Selection with a Cardinality Constraint

**Shinji Ito**
NEC Corporation

**Daisuke Hatano**
RIKEN AIP

**Hanna Sumita**
Tokyo Metropolitan University

**Akihiro Yabe**
NEC Corporation

**Takuro Fukunaga**
RIKEN AIP, JST PRESTO

**Naonori Kakimura**
Keio University

**Ken-ichi Kawarabayashi**
National Institute of Informatics

## Abstract

*Online portfolio selection* is a sequential decision-making problem in which a learner repetitively selects a portfolio over a set of assets, aiming to maximize long-term return. In this paper, we study the problem with the cardinality constraint that the number of assets in a portfolio is restricted to be at most $k$, and consider two scenarios: (i) in the *full-feedback setting*, the learner can observe price relatives (rates of return to cost) for all assets, and (ii) in the *bandit-feedback setting*, the learner can observe price relatives only for invested assets. We propose efficient algorithms for these scenarios, which achieve sublinear regrets. We also provide regret (statistical) lower bounds for both scenarios which nearly match the upper bounds when $k$ is a constant. In addition, we give a computational lower bound, which implies that no algorithm maintains both computational efficiency, as well as a small regret upper bound.

## 1 Introduction

Online portfolio selection [10, 22] is a fundamental problem in financial engineering, in which a learner sequentially selects a portfolio over a set of assets, aiming to maximize cumulative wealth. For this problem, principled algorithms (e.g., the universal portfolio algorithm [10]) have been proposed, which behave as if one knew the empirical distribution of future market performance. On the other hand, these algorithms work only under the strong assumption that we can hold portfolios of arbitrary combinations of assets, and that we can observe price relatives, the multiplicative factors by which prices change, for all assets. Due to these limitations, this framework does not directly apply to such real-world applications as investment in advertising or R&D, where the available combination of assets is restricted and/or price relatives (return on investment) are revealed only for assets that have been invested in.

In order to overcome such issues, we consider the following problem setting: Suppose that there are $T$ rounds and a market has $d$ assets, represented by $[d] := \{1, \ldots, d\}$. In each round $t$, we design a portfolio, that represents the proportion of the current wealth invested in each of the $d$ assets. That is, a *portfolio* can be expressed as a vector $\mathbf{x}_t = [x_{t1}, \ldots, x_{td}]^\top$ such that $x_{ti} \geq 0$ for all $i \in [d]$ and $\sum_{i=1}^d x_{ti} \leq 1$. The combination of assets is restricted with a set of *available combinations* $\mathcal{S} \subseteq 2^{[d]}$, that is, a portfolio $\mathbf{x}_t$ must satisfy $\mathrm{supp}(\mathbf{x}_t) = \{i \in [d] \mid x_{ti} \neq 0\} \in \mathcal{S}$. Thus, in each period $t$, we choose $S_t$ from $\mathcal{S}$ and determine a portfolio $\mathbf{x}_t$ only from assets in $S_t$. A typical example of $\mathcal{S}$ can be given by cardinality constraints, i.e., $\mathcal{S}_k := \{S \subseteq [d] \mid |S| = k\}$ for some $k \leq d$. We denote by $\mathbf{r}_t = [r_{t1}, \ldots, r_{td}]^\top$ a *price relative vector*, where $1 + r_{ti}$ is the price relative for the $i$-th asset in the $t$-th period. Then the wealth $A_T$ resulting from the sequentially rebalanced portfolios $\mathbf{x}_1, \ldots, \mathbf{x}_t$ is given by $A_T = \prod_{t=1}^T (1 + \mathbf{r}_t^\top \mathbf{x}_t)$. The best constant portfolio strategy earns the wealth

Table 1: Regret bounds for the full-feedback setting.

| Constraints | Upper bound by Algorithm 1 | Lower bound |
|---|---|---|
| Single asset ($\mathcal{S} = \mathcal{S}_1$) | $R_T = O(\sqrt{T \log d})$ | $R_T = \Omega(\sqrt{T \log d})$ |
| Combination ($\mathcal{S} = \mathcal{S}_k$) | $R_T = O\left(\sqrt{Tk \log \frac{d}{k}}\right)$ <br> ( run in $T\binom{d}{k}\mathrm{poly}(k)$-time ) | $R_T = \Omega\left(\sqrt{T \log \frac{d}{k}}\right)$ for $d \geq 17k$ <br> and no $\mathrm{poly}(d,k,T)$-time algorithm achieves $R_T \leq T^{1-\delta}\mathrm{poly}(d,k)$ |

Table 2: Regret bounds for the bandit-feedback setting.

| Constraint | Upper bound by Algorithm 2 | Lower bound |
|---|---|---|
| Single asset ($\mathcal{S} = \mathcal{S}_1$) | $R_T = O(\sqrt{dT \log T})$ | $R_T = \Omega(\sqrt{dT})$ |
| Combination ($\mathcal{S} = \mathcal{S}_k$) | $R_T = O\left(\sqrt{Tk\binom{d}{k} \log T}\right)$ <br> ( run in $T\mathrm{poly}(d,k)$-time ) | $R_T = \Omega\left(\sqrt{T \left(\frac{d}{Ck^3}\right)^k}\right)$ for $d > k$ <br> and no $\mathrm{poly}(d,k,T)$-time algorithm achieves $R_T \leq T^{1-\delta}\mathrm{poly}(d,k)$ |

$A_T^* := \max_{\mathbf{x}} \prod_{t=1}^T (1 + \mathbf{r}_t^\top \mathbf{x})$ subject to the constraint that $\mathbf{x}$ is a portfolio satisfying $\mathrm{supp}(\mathbf{x}) \in \mathcal{S}$. The performance of our portfolio selection is measured by $R_T = \log A_T^* - \log A_T$, which we call *regret*. The reason that we use $\log A_T$ rather than $A_T$ comes from capital growth theory [16, 21].[1] In terms of the observable information, we consider two different settings: (i) in the *full-feedback setting*, we can observe all the price relatives $r_{ti}$ for $i = 1, \ldots, d$, and (ii) in the *bandit-feedback setting*, we can observe the price relatives $r_{ti}$ only for $i \in S_t$. Note that in each round $t$ a portfolio $\mathbf{x}_t$ has to be determined before knowing $r_{ti}$ in either of the settings. Note also that we do not make any statistical assumption about the behavior of $r_{ti}$, but we assume that $r_{ti}$ is bounded in a closed interval $[C_1, C_2]$, where $C_1$ and $C_2$ are constants satisfying $-1 < C_1 \leq C_2$.

Our problem is a generalization of the standard online portfolio selection problem. In fact, if portfolios combining all assets are available, i.e., if $\mathcal{S} = 2^{[d]}$, then our problem coincides with the standard online portfolio selection problem. For this special case, it has been shown that some online convex optimization (OCO) methods [18, 17, 27] (e.g., the online Newton step method) achieve regret of $O(d \log T)$, and that any algorithm will suffer from regret of $\Omega(d \log T)$ in the worst case [26].

Our contribution is twofold; algorithms with sublinear regret upper bounds, and analyses proving regret lower bounds. First, we propose the following two algorithms:

- Algorithm 1 for the full-feedback setting, achieving regret of $O(\sqrt{T \log |\mathcal{S}|})$.
- Algorithm 2 for the bandit-feedback setting, achieving regret of $O(\sqrt{Tk|\mathcal{S}| \log T})$, where $k$ denotes the largest cardinality among elements in $\mathcal{S}$, i.e., $k = \max_{S \in \mathcal{S}} |S|$.

Tables 1 and 2 summarize the regret bounds for the special case in which the cardinality of assets is restricted to be at most 1 or at most $k$. As shown in Table 1, Algorithm 1 can achieve regret of $O(\sqrt{T}\mathrm{poly}(d))$ even if $k = \Omega(d)$ when $\mathcal{S}$ has an exponentially large size with respect to $d$. In such a case, however, Algorithm 1 requires exponentially large computational time. For the bandit-feedback setting, the regret upper bound can be exponential w.r.t. $d$ if $k = \Omega(d)$, but it is still sublinear in $T$. One main idea behind our algorithms is to combine the multiplicative weight update method (MWU) [3, 14] (in the full-feedback setting) / multi-armed bandit algorithms (MAB) [5, 6] (in the bandit-feedback setting) with OCO. Specifically, for choosing the combination $S_t$ of assets, we employ MWU/MAB, which are online decision making methods over a finite set of actions. For maintaining the proportion $\mathbf{x}_t$ of portfolios, we use OCO, that is, online decision making methods for convex objectives over a convex set of actions.

Second, we show regret lower bounds for both the full-feedback setting and the bandit-feedback setting where $\mathcal{S} = \mathcal{S}_k$, which give insight into the tightness of regret upper bounds achieved with our algorithms. As shown in Table 1, the proven lower bounds for the full-feedback setting are tight up to the $O(\sqrt{k})$ term. For the bandit-feedback setting, the lower bounds are also tight up to the $O(\sqrt{\log T})$ term, if $k = O(1)$. Note that, if $k = d$ then the problem coincides with the standard online portfolio

selection problem, and hence, there exist algorithms achieving $R_T = O(\sqrt{T \log d})$. This implies that the assumption of $d = \Omega(k)$ is essential for proving the lower bounds of $\Omega(\sqrt{T})$. We also note that these *statistical* lower bounds are valid for arbitrary learners, including exponential-time algorithms. Besides statistical ones, we also show *computational* lower bounds suggesting that there is no polynomial-time algorithm achieving a regret bound with a sublinear term in $T$ and a polynomial term w.r.t. $d$ and $k$, unless $\mathbf{NP} \subseteq \mathbf{BPP}$. This means that we cannot improve the computational efficiency of Algorithm 1 to $O(\mathrm{poly}(d, k, T))$-time while preserving its regret upper bound.

To prove the regret lower bounds, we use three different techniques: for the statistical lower bound for the full-feedback setting, we consider a completely random market and evaluate how well the "best" strategy worked after observing the market behavior, in a similar way to that for the lower bound for MWU [3]; for the bandit-feedback setting, we construct a "good" combination $S^* \in \mathcal{S}$ of assets so that it is hard to distinguish it from the others, and bound the number of choosing this "good" combination via a technique similar to that used in the proof of the regret lower bound for MAB [5]; to prove the computational lower bound, we reduce the 3-dimensional matching problem (3DM), one of Karp's 21 NP-complete problems [20], to our problem.

## 2 Related work

Online portfolio selection has been studied in many research areas, including finance, statistics, machine learning, and optimization [1, 10, 19, 22, 23] since Cover [10] formulated the problem setting and proposed a *universal portfolio algorithm* that achieves regret of $O(d \log T)$ with exponential computation cost. This regret upper bound was shown to be optimal by Ordentlich and Cover [26]. The computation cost was reduced by the celebrated work on the online gradient method of Zinkervich [29] for solving *online convex optimization* (OCO) [17, 27], a general framework including online portfolio selection, but the regret bound is $O(d\sqrt{T})$ and suboptimal for online portfolio selection. A breakthrough w.r.t. this suboptimality came with the *online Newton step* and the *follow-the-approximation-leader* method of Hazan et al. [18], which are computationally efficient and achieve regret of $O(d \log T)$ for a special case of OCO, including online portfolio selection. Among studies on online portfolio selection, the work by Das et al. [12] has a motivation similar to ours: the aim of selecting portfolios with a group-sparse structure. However, their problem setting differs from ours in that they did not put constraints about sparsity but, rather, defined regret containing regularizer inducing group sparsity, and that they supposed that a learner can observe price relatives for all assets after determining portfolios. In contrast to this, our work deals with the sparsity constraint on portfolios, and our methods work even for the bandit-feedback setting, in which feedbacks are observed only on assets that have been invested in.

Another closely related topic is the multi-armed bandit problem (MAB) [4, 5, 6]. For nonstochastic MAB problems, a nearly optimal regret bound is achieved by the Exp3 algorithm [5], which our algorithm strongly relies on. For combinatorial bandit problems [7, 8, 9] in which each arm corresponds to a subset, the work by Chen et al. [8] gives solutions to a wide range of problems. However, this work does not directly apply to our setting, because we need to maintain not only subsets $S_t$ but also continuous variables $\mathbf{x}_t$, and both of them affect regret.

## 3 Upper bounds

### 3.1 Notation and preliminary consideration

Let us introduce some notations. For $S \subseteq [d]$, denote by $\Delta^S$ the set of portfolios whose supports are included in $S$, i.e., $\Delta^S = \left\{ \mathbf{x} \mid x_i \geq 0 \ (i \in [d]), \sum_{i=1}^{d} x_i \leq 1, \mathrm{supp}(\mathbf{x}) \subseteq S \right\}$. Let $(S^*, \mathbf{x}^*)$ denote the optimal fixed strategy for $T$ rounds, i.e., $(S^*, \mathbf{x}^*) \in \underset{S \in \mathcal{S}, \mathbf{x} \in \Delta^S}{\arg \max} \sum_{t=1}^{T} \log(1 + \mathbf{r}_t^\top \mathbf{x})$. Let $\mathbf{x}_t$ denote the output of an algorithm for the $t$-th round. Then the regret $R_T$ of the algorithm can be expressed as

$$R_T = \max_{S \in \mathcal{S}, \mathbf{x} \in \Delta^S} \sum_{t=1}^{T} \log(1 + \mathbf{r}_t^\top \mathbf{x}) - \sum_{t=1}^{T} \log(1 + \mathbf{r}_t^\top \mathbf{x}_t) = \sum_{t=1}^{T} \log(1 + \mathbf{r}_t^\top \mathbf{x}^*) - \sum_{t=1}^{T} \log(1 + \mathbf{r}_t^\top \mathbf{x}_t).$$

---

**Algorithm 1** An algorithm for the full-feedback setting.

---

**Input:** The number $T$ of rounds. The number $d$ of assets. The set of available subsets $\mathcal{S} \subseteq 2^{[d]}$.
    Parameters $\eta > 0$ and $\beta > 0$.
1: Set $\mathbf{w}_1 = (w_1^S)_{S \in \mathcal{S}} \in \mathbb{R}^{\mathcal{S}}$ and $(\mathbf{x}_1^S)_{S \in \mathcal{S}}$ by $w_1^S = 1$ and $\mathbf{x}_1^S = \mathbf{0}$, respectively, for $S \in \mathcal{S}$.
2: **for** $t = 1, \ldots, T$ **do**
3:     Set $S_t$ by randomly choosing $S \in \mathcal{S}$ with a probability proportional to $w_t^S$, i.e., choose $S$ with
       probability $w_t^S / \|\mathbf{w}_t\|_1$.
4:     Output $S_t$ and $\mathbf{x}_t = \mathbf{x}_t^{S_t}$ and observe $r_{ti}$ for all $i \in [d]$.
5:     Update $\mathbf{w}_t$; set $\mathbf{w}_{t+1}$ by $w_{t+1}^S = w_t^S (1 + \mathbf{r}_t^\top \mathbf{x}_t^S)^\eta$ for $S \in \mathcal{S}$.
6:     Update $\mathbf{x}_t^S$; set $\mathbf{x}_{t+1}^S$ by equation (3) for $S \in \mathcal{S}$.
7: **end for**

---

The algorithms presented in this section maintain vectors $\mathbf{x}_t^S \in \Delta^S$ for all $S \in \mathcal{S}$ at the beginning of the $t$-th round. They then choose $S_t$ from $\mathcal{S}$, and output $(S_t, \mathbf{x}_t^{S_t})$. Although other vectors $\mathbf{x}_t^S$ ($S \neq S_t$) do not appear in the output, they are used to compute outputs in subsequent rounds.

In the computation of $\mathbf{x}_{t+1}^S$, we refer to the following vectors $\mathbf{g}_t$ and matrices $H_t^S$:

$$\mathbf{g}_t^S = \frac{\mathbf{r}_t|_S}{1 + \mathbf{r}_t^\top \mathbf{x}_t^S}, \quad H_t^S = \frac{(1 + C_1)^2}{(1 + C_2)^2} \mathbf{g}_t^S \mathbf{g}_t^{S\top} = C_3 \mathbf{g}_t^S \mathbf{g}_t^{S\top}, \tag{1}$$

where $\mathbf{r}_t|_S = [r'_{t1}, \ldots, r'_{td}]^\top$ is defined by $r'_{ti} = r_{ti}$ for $i \in S$ and $r'_{ti} = 0$ for $i \in [d] \setminus S$. These $\mathbf{g}_t^S$ and $H_t^S$ have the following property which plays an important role in our analysis:

**Lemma 1.** *For any* $\mathbf{x} \in \Delta^S$*, it holds that*

$$\log(1 + \mathbf{r}_t^\top \mathbf{x}) - \log(1 + \mathbf{r}_t^\top \mathbf{x}_t^S) \leq \mathbf{g}_t^{S\top}(\mathbf{x} - \mathbf{x}_t^S) - \frac{1}{2}(\mathbf{x} - \mathbf{x}_t^S)^\top H_t^S (\mathbf{x} - \mathbf{x}_t^S). \tag{2}$$

For the proof, see Appendix B in the supplementary material.

### 3.2 Algorithm for the full-feedback setting

We propose an algorithm for the full-feedback setting, created by combining the multiplicative weight update method (MWU) [3] and the follow-the-approximate-leader method (FTAL) [18]. More specifically, our proposed algorithm updates the probability of choosing a subset $S \in \mathcal{S}$ by MWU and updates the portfolio vector $\mathbf{x}_t^S$ by FTAL. The entire algorithm is summarized in Algorithm 1.

Our algorithm maintains *weight* $w_t^S \geq 0$ and a portfolio vector $\mathbf{x}_t^S$ for each subset $S \in \mathcal{S}$ at the begining of the $t$-th round, where $w_1^S$ and $\mathbf{x}_1^S$ are initialized by $w_1^S = 1$ and $\mathbf{x}_1^S = \mathbf{0}$ for all $S \in \mathcal{S}$. In each round $t$, a subset $S_t$ is chosen with a probability proportional to $w_t^S$. Given the feedback $\mathbf{r}_t$, the algorithm computes $w_{t+1}^S$ and $\mathbf{x}_{t+1}^S$. The weight $w_{t+1}^S$ is obtained from $w_t^S$ by multiplying $(1 + \mathbf{r}_t^\top \mathbf{x}_t^S)^\eta$, where $\eta > 0$ is a parameter we optimize later. The portfolio vector $\mathbf{x}_{t+1}^S$ is computed by FTAL as follows:

$$\mathbf{x}_{t+1}^S \in \arg\max_{\mathbf{x} \in \Delta^S} \left\{ \sum_{j=1}^t \left( \mathbf{g}_j^{S\top}(\mathbf{x} - \mathbf{x}_j^S) - \frac{1}{2}(\mathbf{x} - \mathbf{x}_j^S)^\top H_j^S (\mathbf{x} - \mathbf{x}_j^S) \right) - \frac{\beta}{2}\|\mathbf{x}\|_2^2 \right\}, \tag{3}$$

where $\beta$ is a regularization parameter optimized later, and $\|\cdot\|$ stands for the $\ell_2$ norm: $\|[x_1, \ldots, x_d]^\top\|_2^2 = \sum_{i=1}^d x_i^2$. Since (3) is a convex quadratic programming problem with linear constraints, $\mathbf{x}_{t+1}^S$ can be computed efficiently by, e.g., interior point methods [24]. Recently, Ye et al. [28] have proposed a more efficient algorithm for solving (3). For the special case of the single asset selection setting, i.e., if $\mathcal{S} = \mathcal{S}_1 = \{\{i\} \mid i \in [d]\}$, then $\mathbf{x}_{t+1}^{\{i\}} = (0, \ldots, 0, x_{t+1,i}, 0, \ldots, 0)$ has a closed-form expression: $x_{t+1,i} = \pi_{[0,1]}\left( \frac{\sum_{j=1}^t g_{ji}}{\beta + C_3 \sum_{j=1}^t g_{ji}^2} \right)$, where $g_{ji} := \frac{r_{ji}}{1 + r_{ji} x_{ji}}$ and $\pi_{[0,1]}(\cdot)$ stands for a projection onto $[0,1]$ defined by $\pi_{[0,1]}(y) = 0$ for $y < 0$, $\pi_{[0,1]}(y) = y$ for $0 \leq y \leq 1$, and $\pi_{[0,1]}(y) = 1$ for $y > 1$.

Our algorithm achieves the regret described below for arbitrary inputs, where constants $C_3, C_4, C_5$ are given by $C_3 = \frac{(1+C_1)^2}{(1+C_2)^2}$, $C_4 = \log \frac{1+C_2}{1+C_1}$, and $C_5 = \frac{\max\{C_1^2, C_2^2\}}{(1+C_1)^2}$.

**Algorithm 2** An algorithm for the bandit-feedback setting.

---

**Input:** The number $T$ of rounds. The number $d$ of assets. The set of available subsets $\mathcal{S} \subseteq 2^{[d]}$.
   Parameters $\eta > 0$, $\gamma \in (0, 1)$ and $\beta > 0$.
1: Set $\mathbf{w}_1 = (w_1^S)_{S \in \mathcal{S}} \in \mathbb{R}^{\mathcal{S}}$ and $(\mathbf{x}_1^S)_{S \in \mathcal{S}}$ by $w_1^S = 1$ and $\mathbf{x}_1^S = \mathbf{0}$, respectively, for $S \in \mathcal{S}$.
2: **for** $t = 1, \ldots, T$ **do**
3:    Set the probability vector $\mathbf{p}_t = (p_t^S)_{S \in \mathcal{S}} \in [0, 1]^{\mathcal{S}}$ by $p_t^S = \frac{\gamma}{|\mathcal{S}|} + (1 - \gamma)\frac{w_t^S}{\|\mathbf{w}_t\|_1}$.
4:    Randomly choose $S_t \in \mathcal{S}$ on the basis of the probability vector $\mathbf{p}_t$.
5:    Output $S_t$ and $\mathbf{x}_t = \mathbf{x}_t^{S_t}$, and observe $r_{ti}$ for $i \in S_t$.
6:    Update $\mathbf{w}_t$; set $\mathbf{w}_{t+1}^{S_t}$ by $w_{t+1}^{S_t} = w_{ti_t}\left(\frac{1 + \mathbf{r}_t^\top \mathbf{x}_t}{1 + C_1}\right)^{\eta/p_{ti_t}}$ and $w_{t+1}^S = w_t^S$ for $S \in \mathcal{S} \setminus \{S_t\}$.
7:    Update $\mathbf{x}_t^S$; set $\mathbf{x}_{t+1}^S$ by equation (7).
8: **end for**

---

**Theorem 2.** *Algorithm 1 achieves the following regret upper bound if $\eta \leq 1/C_4$:*

$$\mathbf{E}[R_T] \leq \frac{\log|\mathcal{S}|}{\eta} + C_4^2 \eta T + \frac{1}{2}\beta + \frac{k}{C_3}\log\left(1 + \frac{C_3 C_5 T}{\beta}\right). \tag{4}$$

*In particular, setting $\eta = \frac{1}{C_4}\min\left\{1, \sqrt{\frac{\log|\mathcal{S}|}{T}}\right\}$ and $\beta = 1$, we obtain*

$$\mathbf{E}[R_T] = O\left(\sqrt{T\log|\mathcal{S}|} + k\log T + \log|\mathcal{S}|\right). \tag{5}$$

**Running time**    If (3) can be computed in $p(k)$-time, Algorithm 1 runs in $O(|\mathcal{S}|p(k))$-time per round. If $\mathcal{S}$ is an exponentially large set, e.g., if $\mathcal{S} = \{S \subseteq [d] \mid |S| = k\}$ and $k = \Theta(d)$, the computational time for $O(|\mathcal{S}|p(k))$ will be exponentially large w.r.t. $d$. This computational complexity is shown to be inevitable in Section 4.1. For the special case of the single asset selection setting, i.e., if $\mathcal{S} = \mathcal{S}_1 = \{\{i\} \mid i \in [d]\}$, Algorithm 1 runs in $O(d)$-time per round since each $\mathbf{x}_t^{\{i\}}$ can be updated in constant time.

### 3.3   Algorithm for the bandit-feedback setting

We construct an algorithm for the bandit-feedback setting by combining the Exp3 algorithm [5] for the multi-armed bandit problem and FTAL. Similarly to the process used in Algorithm 1, the algorithm updates the probability of choosing $S_t \in \mathcal{S}$ by the Exp3 algorithm (in place of MWU) and updates portfolios $\mathbf{x}_t^S$ by FTAL. The main difficulty comes from the fact that the learner cannot observe all the entries of $(r_{ti})_{i=1}^d$. Due to this limitation, we cannot always update $\mathbf{x}_t^S$ for all $S \in \mathcal{S}$. In order to deal with this problem, we construct unbiased estimators of $\mathbf{g}_t^S$ and $H_t^S$ for each $S \in \mathcal{S}$ by

$$\hat{\mathbf{g}}_t^{S_t} = \frac{\mathbf{g}_t^{S_t}}{p_t^{S_t}}, \quad \hat{H}_t^{S_t} = \frac{H_t^{S_t}}{p_t^{S_t}}, \qquad \hat{\mathbf{g}}_t^S = \mathbf{0}, \quad \hat{H}_t^S = O \quad (S \in \mathcal{S} \setminus \{S_t\}), \tag{6}$$

where $p_t^S$ is the probability of choosing $S$ in round $t$, which is computed by a procedure similar to that used in the Exp3 algorithm. Note that $\hat{\mathbf{g}}_t^S$ and $\hat{H}_t^S$ can be calculated from the observed information alone. Using these unbiased estimators, we compute the portfolio vectors $\mathbf{x}_{t+1}^S$ by FTAL as follows:

$$\mathbf{x}_{t+1}^S \in \arg\max_{\mathbf{x} \in \Delta^S}\left\{\sum_{j=1}^t\left(\hat{\mathbf{g}}_j^{S\top}(\mathbf{x} - \mathbf{x}_j^S) - \frac{1}{2}(\mathbf{x} - \mathbf{x}_j^S)^\top \hat{H}_j^S(\mathbf{x} - \mathbf{x}_j^S)\right) - \frac{1}{2}\beta\|\mathbf{x}\|_2^2\right\}. \tag{7}$$

Note that $\mathbf{x}_{t+1}^S = \mathbf{x}_t^S$ for each $S \in \mathcal{S} \setminus \{S_t\}$ since $\hat{\mathbf{g}}_t^S = \mathbf{0}$ and $\hat{H}_t^S = O$. Hence the convex quadratic programming problem (7) is solved only once in each round. The entire algorithm is summarized in Algorithm 2.

**Theorem 3.** *Algorithm 2 achieves the following regret upper bound if $\eta \leq \frac{\gamma}{C_4|\mathcal{S}|}$:*

$$\mathbf{E}[R_T] \leq \frac{\log|\mathcal{S}|}{\eta} + (C_4^2 \eta|\mathcal{S}| + C_4\gamma)T + \frac{1}{2}\beta + \frac{k|\mathcal{S}|}{C_3\gamma}\log\left(1 + \frac{C_3 C_5 T}{\beta}\right). \tag{8}$$

*Setting $\gamma = \min\left\{1, \sqrt{\frac{k|\mathcal{S}|\log(1+T)}{T}}\right\}$, $\eta = \frac{\gamma}{C_4|\mathcal{S}|}\min\left\{1, \sqrt{\frac{\log|\mathcal{S}|}{k\log(1+T)}}\right\}$ and $\beta = C_3 C_5$, we obtain*

$$\mathbf{E}[R_T] = O\left(\sqrt{T|\mathcal{S}|k\log T} + |\mathcal{S}|\sqrt{k\log|\mathcal{S}|\log T} + |\mathcal{S}|k\right).$$

**Running time**  Algorithm 2 runs in $O(p(k) + \log^2(|\mathcal{S}|))$-time per round, assuming that (7) can be computed in $p(k)$-time. In fact, from the definition (6) of $\hat{\mathbf{g}}_t^S$ and $\hat{H}_t^S$, the update of $\mathbf{x}_t^S$ given by (7) is needed only for $S = S_t$. Furthermore, for $\mathcal{S} = \{S_1, S_2, \ldots, S_{|\mathcal{S}|}\}$, both updating $w_t^S$ for some $S \in \mathcal{S}$ and computing the prefix sum $\sum_{j=1}^{i} w_t^{S_j}$ for some $i \in [|\mathcal{S}|]$ can be performed in $O(\log|\mathcal{S}|)$-time by using a Fenwick tree [13]. This implies that sampling $S_t$ w.r.t. $p_t^S = \frac{\gamma}{|\mathcal{S}|} + \frac{w_t^S}{\|\mathbf{w}_t\|^S}$ can be performed in $O(\log^2|\mathcal{S}|)$-time.

# 4 Lower bounds

In this section, we present lower bounds on regrets achievable by algorithms for the online portfolio selection problem. We focus on the case of $\mathcal{S} = \mathcal{S}_k = \{S \subseteq [d] \mid |S| = k\}$ throughout this section.

## 4.1 Computational complexity

We show that, unless the complexity class **BPP** includes **NP**, there exists no algorithm for the online problem with a cardinality constraint such that its running time is polynomial both on $d$ and $T$ and its regret is bounded by a polynomial in $d$ and sublinear in $T$. This fact is shown by presenting a reduction from the 3-dimensional matching problem (**3DM**). An instance $U$ of **3DM** consists of 3-tuples $(x_1, y_1, z_1), \ldots, (x_d, y_d, z_d) \in [k] \times [k] \times [k]$. Two tuples, $(x_i, y_i, z_i)$ and $(x_j, y_j, z_j)$, are called disjoint if $x_i \neq x_j$, $y_i \neq y_j$, and $z_i \neq z_j$. The task of **3DM** is to determine whether or not there exist $k$ pairwise-disjoint tuples; if they do exist, we write $U \in$ **3DM**.

From a **3DM** instance $U = \{(x_j, y_j, z_j)\}_{j=1}^{d}$, we construct an input sequence $(\mathbf{r}_t)_{t=1,\ldots,T}$ of the online portfolio selection problem as follows. Let $A = (a_{ij}) \in \{0,1\}^{3k \times d}$ be a matrix such that $a_{ij} = 1$ if $i = x_j$ or $i = k + y_j$ or $i = 2k + z_j$, and $a_{ij} = 0$ otherwise. From $A$, we construct $B \in \mathbb{R}^{3k \times (d+1)}$ by $B = \frac{1}{3k}[A, -\mathbf{1}_{3k}]$, where $\mathbf{1}_{3k}$ is the all-one vector of dimension $3k$. Let $T \geq \max\{(4 \cdot 5184k^4)^2, (5184k^4 \cdot p_2(d))^{\frac{1}{\delta}}\}$ for an arbitrary polynomial $p_2$ and an arbitrary positive parameter $\delta$. For each $t \in [T]$, take $\mathbf{z}_t$ from the uniform random distribution on $\{-1, 1\}^{3k}$, independently. Then, $\mathbf{r}_t$ can be defined by $\mathbf{r}_t = \mathbf{1}_{d+1} + B^\top \mathbf{z}_t$ for each $t \in [T]$. Note that $\mathbf{r}_t \in [0, 2]^{(d+1)}$ holds for each $t \in [T]$.

We give the sequence $(\mathbf{r}_t)_{t=1,\ldots,T}$ to an algorithm $\mathcal{A}$. Let $(\mathbf{x}_t)_{t=1,\ldots,T}$ denote the sequence output by $\mathcal{A}$. We determine that $U \in$ **3DM** if $\sum_{t=1}^{T} \log(1 + \mathbf{r}_t^\top \mathbf{x}_t) \geq T(\log 2 - \frac{1}{5184k^4})$ holds, while otherwise we determine $U \notin$ **3DM** to hold. We can prove that this determination is correct with a probability of at least $2/3$. For the proof, see Appendix E in the supplementary material.

**Theorem 4.** *Let $\delta$ be an arbitrary positive number, and $p_1$ and $p_2$ be arbitrary polynomials. Assume that there exists a $p_1(d, T)$-time algorithm $\mathcal{A}$ for the full-feedback online portfolio selection problem with $\mathcal{S} = \mathcal{S}_{k+1}$ that achieves regret $R_T \leq p_2(d)T^{1-\delta}$ with a probability of at least $2/3$. Then, given a **3DM** instance $U \subseteq [k] \times [k] \times [k]$, one can decide if $U \in$ **3DM** with a probability of at least $2/3$ in $p_1(|U|, \max\{k^8, (k^4 p_2(|U|))^{\frac{1}{\delta}}\})$-time.*

**Corollary 5.** *Under the assumption of* **NP** $\not\subseteq$ **BPP**, *if an algorithm achieves $O(p(d,k)T^{1-\delta})$ regret for arbitrary $d$ and arbitrary $k$, the algorithm will not run in polynomial time, i.e., the running time will be larger than any polynomial for some $d$ and some $k$.*

Note that the computational lower bounds described in Theorem 4 and Corollary 5 are also valid for the bandit-feedback setting, since algorithms for the bandit-feedback settings can be used for the full-feedback setting.

## 4.2 Regret lower bound for the full-feedback setting

We show here that, for the full-feedback setting of the online portfolio selection problem with $\mathcal{S} = \mathcal{S}_k$, every algorithm (including exponential-time algorithms) suffers from regret of $\Omega\left(\sqrt{T \log \frac{d}{k}}\right)$ in the worst case. We can show this by analyzing the behavior of an algorithm for a certain random input. In the analysis, we use the fact that the following two inequalities hold when $\mathbf{r_t}$ follows the discrete uniform distribution on $\{0, 1\}^d$ independently:

$$\underset{\mathbf{r}_t,\mathbf{x}_t}{\mathbf{E}}\left[\sum_{t=1}^{T}\log(1+\mathbf{r}_t^\top\mathbf{x}_t)\right]\leq T\underset{X}{\mathbf{E}}\left[\log\left(1+\frac{1}{k}X\right)\right],$$

$$\underset{\mathbf{r}_t,\mathbf{x}_t}{\mathbf{E}}\left[\max_{S\in\mathcal{S}_k,\mathbf{x}\in\Delta^S}\sum_{t=1}^{T}\log(1+\mathbf{r}_t^\top\mathbf{x})\right]\geq T\cdot\underset{X}{\mathbf{E}}\left[\log\left(1+\frac{1}{k}X\right)\right]+\Omega\left(\sqrt{T\log\frac{d}{k}}\right),$$

where $X$ is a binomial random variable following $B(k,1/2)$. See Appendix F for details regarding the proof.

**Theorem 6.** *Let $d\geq 17k$, and consider the online portfolio selection problem with $d$ assets and available combinations $\mathcal{S}=\mathcal{S}_k$. There is a probability distribution of input sequences $\{\mathbf{r}_t\}_{t=1}^T$ such that the regret of any algorithm for the full-feedback setting is bounded as $\mathbf{E}[R_T]=\Omega\left(\sqrt{T\log\frac{d}{k}}\right)$, where the expectation is with respect to the randomness of both $\mathbf{r}$ and the algorithm.*

### 4.3 Regret lower bound for the bandit-feedback setting

In this subsection, we consider the bandit-feedback setting of the online portfolio selection problem with $\mathcal{S}=\mathcal{S}_k$. We show that every algorithm (including exponential-time algorithms) for this setting suffers from regret of $\Omega\left(\sqrt{T(\frac{d}{Ck^3})^k}\right)$ when the input sequence is defined as follows. Let $S^*\in\mathcal{S}_k$. We define a random distribution $D_{S^*}$ on $\{-1,1\}^d$ so that a random vector $\mathbf{z}=[z_1,\ldots,z_d]^\top$ following this distribution satisfies

$$\prod_{i\in S^*}z_i=\begin{cases}1 & \text{w.p. }1/2-\epsilon\\-1 & \text{w.p. }1/2+\epsilon\end{cases},\quad \prod_{i\in S}z_i=\begin{cases}1 & \text{w.p. }1/2\\-1 & \text{w.p. }1/2\end{cases}\quad(S\in 2^{[d]}\setminus\{\emptyset,S^*\}).$$

Such a distribution can be constructed as follows: fix an index $i^*\in S^*$, let $z_i=\begin{cases}1 & \text{w.p. }1/2\\-1 & \text{w.p. }1/2\end{cases}$ for each $i\in[d]\setminus\{i^*\}$, and let $z_0=\begin{cases}1 & \text{w.p. }1/2-\epsilon\\-1 & \text{w.p. }1/2+\epsilon\end{cases}$ independently. Define $z_{i^*}=z_0\prod_{i\in S^*\setminus\{i^*\}}z_i$. Then $\mathbf{z}=[z_1,\ldots,z_d]^\top\sim D_{S^*}$. The price relative vector $\mathbf{r}_t$ in the $t$-th round can be defined by $\mathbf{r}_t=\mathbf{1}_d-\mathbf{z}_t$, where $\mathbf{z}_t\sim D_S^*$ independently for $t\in[T]$. We can show that $\mathbf{r}_t|_S$ follows a uniform distribution for any $S\in\mathcal{S}_k\setminus\{S^*\}$ and only $\mathbf{r}_t|_{S^*}$ follows a slightly different distribution. Because of this, it is difficult for algorithms to distinguish $S^*$ from others, which makes their regrets large. For more details, see Appendix G.

**Theorem 7.** *Let $d\geq k-1$, and consider the online portfolio selection problem with $d$ assets and available combinations $\mathcal{S}=\mathcal{S}_k$. There is a probability distribution of input sequences $\{\mathbf{r}_t\}_{t=1}^T$ such that the regret of any algorithm for the bandit-feedback setting is bounded as $\mathbf{E}[R_T]=\Omega\left(\min\left\{\frac{T}{k(Ck)^k},\sqrt{T(\frac{d}{Ck^3})^k}\right\}\right)$, where the expectation is with respect to the randomness of both $\mathbf{r}$ and the algorithm, and $C$ is a constant depending on $C_1$ and $C_2$.*

## 5 Experimental evaluation

We show the empirical performance of our algorithms through experiments over synthetic and real-world data. In this section, we consider the online portfolio selection problem with $\mathcal{S}=\mathcal{S}_1$. A problem instance is parameterized by a tuple $(d,T,\{\mathbf{r}_t\}_{t=1}^T)$. A synthetic instance is generated as follows: given parameters $d$, $T$, $C_1$, and $C_2$, we randomly choose an asset $i^*$ from $[d]$, and generate $r_{ti^*}\sim U((C_2+C_1)/2,C_2)$ and $r_{ti}\sim U(C_1,C_2)$ for $i\in[d]\setminus\{i^*\}$.

We also conduct our experiments for two real-world instances. The first is based on crypto coin historical data[2], including dates and price data for 19 crypto coins. From this data, we select 7 crypto coins, each having 929 prices, and obtain price relatives $r_{ti}$ of coin $i$ at time $t$ by $(p_{ti}/p_{t-1,i})-1$, where $p_{ti}$ indicates the price of coin $i$ at time $t$. Thus, $d=7$ and $T=928$ in this instance. The other

instance is based on S&P 500 stock data[3], including dates and price data for 505 companies. From this data, we choose $d = 470$ companies, each having 1259 stock prices, and compute $T = 1258$ price relatives for each company in the same way.

For purposes of comparison, we prepare three baseline algorithms: Exp3_cont, Exp3_disc, and MWU_disc. MWU_disc (based on MWU [3]) works in the full-feedback setting and is compared with Algorithm 1. Exp3_cont and Exp3_disc (based on Exp3 [5]) work in the bandit-feedback setting and are compared with Algorithm 2. These baseline algorithms have different ways of updating $\mathbf{x}_t^S$ from those of Algorithms 1 and 2. Note that since $\mathcal{S} = \mathcal{S}_1 = \{\{i\} \mid i \in [d]\}$, $\mathbf{x}_t^S$ can be expressed as $\mathbf{x}_t^S = \mathbf{x}_t^{\{i\}} = [0, \ldots, 0, x_{ti}, 0, \ldots, 0]^\top$. Below, we offer a brief explanation of the comparisons.

**MWU_disc** Set $x_{ti} = 1$ if $\sum_{j=1}^{t-1} r_{ji} \geq 0$ and $x_{ti} = 0$ otherwise. For each $t \in [T]$, select $i_t$ by MWU, where rewards in the $t$-th round are given by $[\log(1 + r_{ti} x_{ti})]_{i=1}^d$, and output $i_t, \mathbf{x}_t^{\{i_t\}}$.

**Exp3_disc** Set $x_{ti} = 1$ if $\sum_{j \in [t-1]:i_j = i} r_{ji} \geq 0$ and $x_{ti} = 0$ otherwise. For each $t \in [T]$, select $i_t$ by Exp3, where reward in the $t$-th round is given by $\log(1 + r_{ti_t} x_{ti_t})$, and output $i_t, \mathbf{x}_t^{\{i_t\}}$.

**Exp3_cont** Set a parameter $B \in \mathbb{N}$, and consider an MAB problem instance with $d(B + 1)$ arms in which the rewards for the $d(B + 1)$ arms in the $t$-th round are given by $(\log(1 + r_{ti} b/B))_{1 \leq i \leq d, 0 \leq b \leq B}$. Apply Exp3 to this MAB problem instance.

We assess the performance of the algorithms on the basis of regrets for synthetic instances and of cumulative price relatives for real-world instances, where regrets and cumulative price relatives are averaged over 10 executions. We set parameters $\eta$ according to Theorem 2 for Algorithm 1 and MWU_disc, and $\eta$ and $\gamma$ according to Theorem 3 for Algorithm 2, Exp3_disc, and Exp3_cont.

Figure 1 shows average regrets for a synthetic instance with $(d, T, C_1, C_2) = (20, 10000, -0.5, 0.5)$. We observe that both Algorithms 1 and 2 converge faster than MWU_disc, Exp3_cont, and Exp3_disc. In addition, the results empirically show that our theoretical bounds are correct.

Figures 2 and 3 show average cumulative price relatives for a real-world instance of S&P 500 stock data with $(d, T, C_1, C_2) = (470, 1258, -0.34, 1.04)$ and for a real-world instance of crypto coin data with $(d, T, C_1, C_2) = (7, 928, -0.7, 3.76)$, respectively. From these figures, we observe that the cumulative price relatives of our algorithms are higher than those of baseline algorithms.

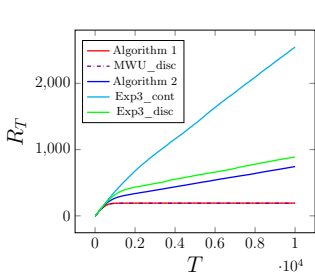
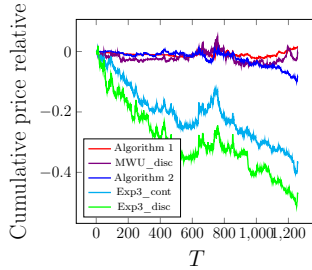
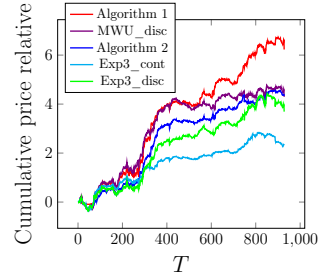

Figure 1: The average regrets over the synthetic dataset with $(d, T, C_1, C_2) = (20, 10000, -0.5, 0.5)$

Figure 2: The average cumulative price relatives over S&P 500 stock dataset

Figure 3: The average cumulative price relatives over the cryptocoin historical dataset

# Acknowledgement

This work was supported by JST ERATO Grant Number JPMJER1201, Japan, and JSPS KAKENHI Grant Number JP18H05291.

## Footnotes

[1] For more details, see Appendix A in the supplementary material.

[2] https://www.kaggle.com/sudalairajkumar/cryptocurrencypricehistory

[3] https://www.kaggle.com/camnugent/sandp500

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
