[Supplementary Material · supplymentary_6757.pdf]

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

# Appendix

## A  A note on the definition of the regret

**Remark 1.** When the reward $A_T$ changes multiplicatively, the expectation of the logarithm $\mathbf{E}[\log A_T]$ can be regarded to be a more reasonable evaluation metrics than would be the expected reward $\mathbf{E}[A_T]$. This is supported by the following example: suppose that $(\mathbf{X}_t)_{t=1}^T = ((X_t^{(1)}, X_t^{(2)}))_{t=1}^T$ are Bernoulli random variables such that $X_t^{(1)} = \begin{cases} 1.3 & \text{w. p. } 0.5 \\ 0.9 & \text{w. p. } 0.5 \end{cases}, X_t^{(2)} = \begin{cases} 2.0 & \text{w. p. } 0.5 \\ 0.4 & \text{w. p. } 0.5 \end{cases},$ and that $\mathbf{X}_t$ and $\mathbf{X}_{t'}$ are independent random variables for $t \neq t'$. Note that we do not assume $X_t^{(1)}$ and $X_t^{(2)}$ to be independent. Define $A_T^{(1)} = \prod_{t=1}^T X_t^{(1)}$ and $A_T^{(2)} = \prod_{t=1}^T X_t^{(2)}$. Then, since $\mathbf{E}[X_t^{(1)}] = 1.1$ and $\mathbf{E}[X_t^{(1)}] = 1.2$, we have $\mathbf{E}[A_T^{(1)}] = 1.1^T < \mathbf{E}[A_T^{(2)}] = 1.2^T$, which implies that we prefer $A_T^{(1)}$ to $A_T^{(2)}$ when determining on the basis of the expectation. However, we can show that $\lim_{T\to\infty} A_T^{(1)} = \infty$ and $\lim_{T\to\infty} A_T^{(2)} = 0$ with probability one, respectively. In fact, if $A_T = \prod_{t=1}^T X_t$ is the product of i.i.d. random variables, we have

$$\lim_{T\to\infty} (A_T)^{\frac{1}{T}} = \exp\left(\lim_{T\to\infty} \frac{1}{T} \sum_{t=1}^T \log X_t\right) = \exp(\mathbf{E}[\log X_1]) \tag{9}$$

with probability one, where the last equality comes from the law of large numbers. Applying (9) to $A_T^{(1)}$ and $A_T^{(2)}$, we obtain $\lim_{T\to\infty}(A_T^{(2)})^{\frac{1}{T}} < 1 < \lim_{T\to\infty}(A_T^{(1)})^{\frac{1}{T}}$ with probability one. In general, if $A_T^{(1)} = \prod_{t=1}^T X_t^{(1)}$ and $A_T^{(2)} = \prod_{t=1}^T X_t^{(2)}$ are products of i.i.d. random variables, then $\mathbf{E}[\log X_1^{(1)}] > \mathbf{E}[\log X_1^{(2)}]$ if and only if $\lim_{t=1}^T A_T^{(1)}/A_T^{(2)} = \infty$ with probability one. These arguments imply that, in the case of a multiplicative reward model, it is reasonable to compare reward logarithms if we focus on events expected to happen with high probability.

## B  Proof of Lemma 1

*Proof.* Since it holds for all $x, x_0 \in [C_1, C_2]$ that $\frac{\mathrm{d}}{\mathrm{d}x}\log(1+x) = \frac{1}{1+x}$ and $\frac{\mathrm{d}^2}{\mathrm{d}x^2}\log(1+x) = -\frac{1}{(1+x)^2}$, we have $\log(1+x) - \log(1+x_0) \leq \frac{x-x_0}{1+x_0} - \frac{(x-x_0)^2}{2(1+C_2)^2} \leq \frac{x-x_0}{1+x_0} - \frac{C_3}{2}\left(\frac{x-x_0}{1+x_0}\right)^2$, where we set $C_3 = \frac{(1+C_1)^2}{(1+C_2)^2}$. Hence, by substituting $x = \mathbf{r}_t^\top \mathbf{x}, x_0 = \mathbf{r}_t^\top \mathbf{x}_t^S$ for arbitrary $t \in [T], S \in \mathcal{S}$ and $\mathbf{x} \in \Delta^S$, we obtain $\log(1 + \mathbf{r}_t^\top \mathbf{x}) - \log(1 + \mathbf{r}_t^\top \mathbf{x}_t^S) \leq \frac{\mathbf{r}_t^\top(\mathbf{x}-\mathbf{x}_t^S)}{1+\mathbf{r}_t^\top \mathbf{x}_t^S} - \frac{C_3}{2}\left(\frac{\mathbf{r}_t^\top(\mathbf{x}-\mathbf{x}_t^S)}{1+\mathbf{r}_t^\top \mathbf{x}_t^S}\right)^2 \leq \mathbf{g}_t^{S\top}(\mathbf{x} - \mathbf{x}_t^S) - \frac{C_3}{2}(\mathbf{g}_t^{S\top}(\mathbf{x}-\mathbf{x}_t^S))^2 = \mathbf{g}_t^{S\top}(\mathbf{x} - \mathbf{x}_t^S) - \frac{1}{2}(\mathbf{x} - \mathbf{x}_t^S)^\top H_t^S(\mathbf{x} - \mathbf{x}_t^S),$  $\square$

## C  Proof of Theorem 2

*Proof.* In the following, we denote $f_t(\mathbf{x}) = \log(1 + \mathbf{r}_t^\top \mathbf{x}) - \log(1 + C_1)$. The regret $R_T$ can be expressed as

$$R_T = \left(\sum_{t=1}^T f_t(\mathbf{x}^*) - \sum_{t=1}^T f_t(\mathbf{x}_t^{S^*})\right) + \left(\sum_{t=1}^T f_t(\mathbf{x}_t^{S^*}) - \sum_{t=1}^T f_t(\mathbf{x}_t^{S_t})\right). \tag{10}$$

Since $S_t$ is chosen by MWU taking the input $(F_t^S)_{S\in\mathcal{S}} = (f_t(\mathbf{x}_t^S))_{S\in\mathcal{S}}$, the second term on the right-hand side of (10) can be bounded as follows (see e.g., [3]):

$$\mathbf{E}\left[\sum_{t=1}^T f_t(\mathbf{x}_t^{S^*}) - \sum_{t=1}^T f_t(\mathbf{x}_t^{S_t})\right] \leq \frac{\log|\mathcal{S}|}{\eta} + C_4^2 \eta T. \tag{11}$$

Since $\mathbf{x}_t^{S^*}$ is computed by FTAL, the first term on the right-hand side of (10) can be bounded as follows (see e.g., [18]):

$$\sum_{t=1}^T f_t(\mathbf{x}^*) - \sum_{t=1}^T f_t(\mathbf{x}_t^{S^*}) \leq \frac{\beta}{2} + \frac{|S^*|}{C_3}\log\left(1 + \frac{C_3 C_5 T}{\beta}\right). \tag{12}$$

Combining (10), (11) and (12), we obtain (4).  $\square$

## D   Proof of Theorem 3

*Proof.* The regret $R_T$ can be expressed as (10). Since $S_t$ is chosen by Exp3 taking the input $(F_t^S)_{S\in\mathcal{S}} = (f_t(\mathbf{x}_t^S))_{S\in\mathcal{S}}$, the second term on the right-hand side of (10) can be bounded as follows (see e.g., [5]):

$$\mathbf{E}\left[\sum_{t=1}^T f_t(\mathbf{x}_t^{S^*}) - \sum_{t=1}^T f_t(\mathbf{x}_t^{S_t})\right] \leq \frac{\log|\mathcal{S}|}{\eta} + (C_4^2\eta|\mathcal{S}| + C_4\gamma)T. \tag{13}$$

The first term on the right-hand side of (10) can be bounded as follows:

$$\mathbf{E}\left[\sum_{t=1}^T f_t(\mathbf{x}^*) - \sum_{t=1}^T f_t(\mathbf{x}_t^{S^*})\right] \leq \mathbf{E}\left[\sum_{t=1}^T (\mathbf{g}_t^{S^*\top}(\mathbf{x}^* - \mathbf{x}_t^{S^*}) - \frac{1}{2}(\mathbf{x}^* - \mathbf{x}_t^{S^*})^\top H_t^{S^*}(\mathbf{x}^* - \mathbf{x}_t^{S^*}))\right]$$

$$= \mathbf{E}\left[\sum_{t=1}^T (\hat{\mathbf{g}}_t^{S^*\top}(\mathbf{x}^* - \mathbf{x}_t^{S^*}) - \frac{1}{2}(\mathbf{x}^* - \mathbf{x}_t^{S^*})^\top \hat{H}_t^{S^*}(\mathbf{x}^* - \mathbf{x}_t^{S^*}))\right], \tag{14}$$

where the inequality comes from (2) and the equality comes from the fact that $\hat{\mathbf{g}}_t^S$ and $\hat{H}_t^S$ are unbiased estimators of $\mathbf{g}_t^S$ and $H_t^S$, respectively. Since $\mathbf{x}_t^{S^*}$ is computed by FTAL as in (7), the right-hand side of can be bounded as follows (see e.g., [18]):

$$\sum_{t=1}^T (\hat{\mathbf{g}}_t^{S^*\top}(\mathbf{x}^* - \mathbf{x}_t^{S^*}) - \frac{1}{2}(\mathbf{x}^* - \mathbf{x}_t^{S^*})^\top \hat{H}_t^{S^*}(\mathbf{x}^* - \mathbf{x}_t^{S^*}))$$

$$\leq \frac{\beta\|\mathbf{x}^*\|_2^2}{2} + \sum_{t=1}^T \hat{\mathbf{g}}_t^{S^*\top}(\beta I + \sum_{j=1}^t \hat{H}_j^{S^*})^{-1}\hat{\mathbf{g}}_t^{S^*}$$

$$\leq \frac{\beta}{2} + \frac{|\mathcal{S}|}{C_3\gamma}\sum_{t=1}^T C_3 p_t^{S^*}\hat{\mathbf{g}}_t^{S^*\top}(\beta I + \sum_{j=1}^t C_3 p_j^{S^*}\hat{\mathbf{g}}_j^{S^*}\hat{\mathbf{g}}_j^{S^*\top})^{-1}\hat{\mathbf{g}}_t^{S^*}$$

$$\leq \frac{\beta}{2} + \frac{|\mathcal{S}|}{C_3\gamma}\log\frac{\det(\beta I + C_3\sum_{j=1}^T p_j^{S^*}\hat{\mathbf{g}}_j^{S^*}\hat{\mathbf{g}}_j^{S^*\top})}{\det \beta I}, \tag{15}$$

where the first and third inequalities come from the standard analysis of FTAL, and the second inequality holds since $p_t^S|\mathcal{S}|/\gamma \leq 1$ from the definition of $p_t^S$. Denote $M_T = C_3\sum_{j=1}^T p_j^{S^*}\hat{\mathbf{g}}_j^{S^*}\hat{\mathbf{g}}_j^{S^*\top}$. Since $\|\mathbf{g}_t^{S^*}\|_0 \leq |S^*| \leq k$, the eigenvalues $\{\lambda_1,\ldots,\lambda_d\}$ of $M_T$ include at least $d-k$ zero eigenvalues. From this and the fact that $\lambda_i \geq 0$ and $\sum_{j=1}^d \lambda_j = \mathrm{tr}(M_t)$, we have $\det(\beta I + M_T) = \prod_{j=1}^d(\beta + \lambda_i) \leq \beta^{d-k}(\beta + \frac{1}{k}\mathrm{tr}(M_T))^k$. This inequality and Jensen's inequality yield $\mathbf{E}[\log(\det(\beta I + M_T))] \leq (d-k)\log\beta + \mathbf{E}[k\log(\beta + \frac{1}{k}\mathrm{tr}(M_T))] \leq (d-k)\log\beta + k\log(\beta + \frac{1}{k}\mathbf{E}[\mathrm{tr}(M_T)])$. Since $\mathbf{E}[\mathrm{tr}(M_T)] = \sum_{t=1}^T \mathbf{E}[\mathrm{tr}(\hat{H}_t^{S^*})] = \sum_{t=1}^T \mathbf{E}[\mathrm{tr}(H_t^{S^*})] \leq TkC_3C_5$, we have $\mathbf{E}[\log(\det(\beta I + M_T))] \leq (d-k)\log\beta + k\log(\beta + TC_3C_5)$. Combining this with (14) and (15), we obtain (8). □

## E   Proof of Theorem 4

*Proof.* From a **3DM** instance $U = \{(x_j, y_j, z_j)\}_{j=1}^d$, we construct an input sequence $(\mathbf{r}_t)_{t=1,\ldots,T}$ for algorithm $\mathcal{A}$ as follows. Let $A = (a_{ij}) \in \{0,1\}^{3k\times d}$ be a matrix such that $a_{ij} = 1$ if $i = x_j$ or $i = k + y_j$ or $i = 2k + z_j$, and $a_{ij} = 0$ otherwise. From $A$, we construct $B \in \mathbb{R}^{3k\times(d+1)}$ by $B = \frac{1}{3k}[A, -\mathbf{1}_{3k}]$, where $\mathbf{1}_{3k}$ is an all-one vector of dimension $3k$. Let $T \geq \max\{(4\cdot 5184k^4)^2, (5184k^4\cdot p_2(d))^{\frac{1}{\delta}}\}$. For each $t \in [T]$, take $\mathbf{z}_t$ from the uniform random distribution on $\{-1,1\}^{3k}$, independently. Then, $\mathbf{r}_t$ can be defined by $\mathbf{r}_t = \mathbf{1}_{d+1} + B^\top\mathbf{z}_t$ for each $t \in [T]$. Note that $\mathbf{r}_t \in [0,2]^{(d+1)}$ holds for each $t \in [T]$.

We give the sequence $(\mathbf{r}_t)_{t=1,\ldots,T}$ to $\mathcal{A}$. Let $(\mathbf{x}_t)_{t=1,\ldots,T}$ denote the sequence output by $\mathcal{A}$. We determine that $U \in \textbf{3DM}$ if $\sum_{t=1}^T \log(1 + \mathbf{r}_t^\top\mathbf{x}_t) \geq T(\log 2 - \frac{1}{5184k^4})$ holds, while otherwise we

determine that $U \notin \textbf{3DM}$ holds. Below, we prove that this determination is correct with a probability of at least $2/3$.

Assume that $U \in \textbf{3DM}$. Then, there exists $\mathbf{y}^* \in \{0,1\}^d$ such that $\|\mathbf{y}^*\|_0 = k$ and $A\mathbf{y}^* = \mathbf{1}_{3k}$, and there exists $\mathbf{y}^* \in \{0,1\}^d$ such that $\|\mathbf{y}^*\|_0 = \|\mathbf{y}^*\|_1 = k$ and $A\mathbf{y}^* = \mathbf{1}_{3k}$. Define $\mathbf{x}^* := \frac{1}{k+1}\begin{bmatrix} \mathbf{y}^* \\ 1 \end{bmatrix}$. The vector $\mathbf{x}^*$ satisfies $\mathbf{x}^* \in \Delta^S$ for some $S \in \mathcal{S}_{k+1}$. Moreover, it holds that $\mathbf{r}_t^\top \mathbf{x}^* = \mathbf{1}_{d+1}^\top \mathbf{x}^* + \mathbf{z}_t^\top B\mathbf{x}^* = 1 + \frac{1}{3k(k+1)}\mathbf{z}_t^\top (A\mathbf{y}^* - \mathbf{1}_{3k}) = 1$. Hence, we obtain

$$\max_{S \in \mathcal{S}_{k+1}, \mathbf{x} \in \Delta^S} \sum_{t=1}^{T} \log(1 + \mathbf{r}_t^\top \mathbf{x}) \geq \sum_{t=1}^{T} \log(1 + \mathbf{r}_t^\top \mathbf{x}^*) = T \log 2.$$

From this inequality and $R_T \leq p_2(d)T^{1-\delta}$ (with a probability $\geq 2/3$), we obtain

$$\sum_{t=1}^{T} \log(1 + \mathbf{r}_t^\top \mathbf{x}_t) \geq \max_{S \in \mathcal{S}_{k+1}, \mathbf{x} \in \Delta^S} \sum_{t=1}^{T} \log(1 + \mathbf{r}_t^\top \mathbf{x}) - R_T$$

$$\geq T \log 2 - p_2(d)T^{1-\delta} \geq T \left( \log 2 - \frac{1}{5184k^4} \right),$$

where the last inequality comes from $T \geq (5184k^4 \cdot p_2(d))^{\frac{1}{\delta}}$. This inequality means that the decision is correct with a probability $\geq 2/3$ if $U \in \textbf{3DM}$.

For the remainder of the proof, we assume that $U \notin \textbf{3DM}$. This assumption implies that, for all $\mathbf{y} \in \mathbb{R}_{\geq 0}^d$ satisfying $\|\mathbf{y}\|_0 \leq k$, we have $\min_{1 \leq i \leq 3k}(A\mathbf{y})_i = 0$. Moreover, since each column of $A$ has at least one entry of value 1, we have $\max_{1 \leq i \leq 3k}(A\mathbf{y})_i \geq \|\mathbf{y}\|_\infty$ for all $\mathbf{y} \in \mathbb{R}_{\geq 0}^d$.

We first prove that $\|B\mathbf{x}\|_2 \geq \frac{1}{12k^2}\|\mathbf{x}\|_1$ holds for all $\mathbf{x} \in \mathbb{R}_{\geq 0}^{d+1}$ satisfying $\|\mathbf{x}\|_0 \leq k+1$. We consider the following two cases: the last entry of $\mathbf{x}$ is either positive or zero. The former case is when $\mathbf{x}$ is expressed as $\mathbf{x} = \begin{bmatrix} \mathbf{y} \\ y_0 \end{bmatrix}$ with $\mathbf{y} \in \mathbb{R}_{\geq 0}^d$, $\|\mathbf{y}\|_0 \leq k$ and $y_0 > 0$. In this case, we have

$$\|B\mathbf{x}\|_\infty \geq \frac{1}{3k} \max \left\{ |\min_{1 \leq i \leq 3k}(A\mathbf{y})_i - y_0|, |\max_{1 \leq i \leq 3k}(A\mathbf{y})_i - y_0| \right\}$$

$$= \frac{1}{3k} \max \left\{ |y_0|, |\max_{1 \leq i \leq 3k}(A\mathbf{y})_i - y_0| \right\}$$

$$\geq \frac{1}{3k} \max \left\{ |y_0|, \frac{1}{2}|\max_{1 \leq i \leq 3k}(A\mathbf{y})_i| \right\} \geq \frac{1}{3k} \max \left\{ |y_0|, \frac{1}{2}\|\mathbf{y}\|_\infty \right\} \geq \frac{1}{6k}\|\mathbf{x}\|_\infty,$$

where the second inequality comes from the fact that arbitrary $y_0$ satisfies $\max\{|y_0|, |a - y_0|\} \geq |a|/2$. In the latter case, namely, when $\mathbf{x} = \begin{bmatrix} \mathbf{y} \\ 0 \end{bmatrix}$ with some $\mathbf{x} \in \mathbb{R}_{\geq 0}^d$ such that $\|\mathbf{y}\|_0 \leq k+1$, we have $\|B\mathbf{x}\|_\infty \geq \frac{1}{3k}\|\mathbf{y}\|_\infty = \frac{1}{3k}\|\mathbf{x}\|_\infty$. Accordingly, in both of these cases, we have $\|B\mathbf{x}\|_\infty \geq \frac{1}{6k}\|\mathbf{x}\|_\infty$, and hence, we have $\|B\mathbf{x}\|_2 \geq \|B\mathbf{x}\|_\infty \geq \frac{1}{6k}\|\mathbf{x}\|_\infty \geq \frac{1}{6k(k+1)}\|\mathbf{x}\|_1 \geq \frac{1}{12k^2}\|\mathbf{x}\|_1$.

Then, since $\log(1+y) \leq \log 2 + \frac{1}{2}(y-1) - \frac{1}{18}(y-1)^2$ for $y \in [0, 2]$, and since $\mathbf{z}_t$ are statistically independent of $\mathbf{x}_t$ and $\mathbf{E}[\mathbf{z}_t] = \mathbf{0}$, $\mathbf{E}[\mathbf{z}_t\mathbf{z}_t^\top] = I$, we have

$$\mathbf{E}_{\mathbf{z}_t, \mathbf{x}_t} [\log(1 + \mathbf{r}_t^\top \mathbf{x}_t)]$$

$$\leq \log 2 + \mathbf{E}_{\mathbf{z}_t, \mathbf{x}_t} \left[ \frac{1}{2}(\|\mathbf{x}_t\|_1 + \mathbf{z}_t^\top B\mathbf{x}_t - 1) - \frac{1}{18}(\|\mathbf{x}_t\|_1 + \mathbf{z}_t^\top B\mathbf{x}_t - 1)^2 \right]$$

$$= \log 2 - \frac{1}{2} + \mathbf{E}_{\mathbf{x}_t, \mathbf{z}_t} \left[ \frac{1}{2}\|\mathbf{x}_t\|_1 - \frac{1}{18} \left( \mathbf{x}_t^\top B^\top \mathbf{z}_t \mathbf{z}_t^\top B\mathbf{x}_t - 2\mathbf{z}_t^\top B\mathbf{x}_t(\|\mathbf{x}_t\|_1 - 1) + (\|\mathbf{x}_t\|_1 - 1)^2 \right) \right]$$

$$\leq \log 2 - \frac{1}{2} + \mathbf{E}_{\mathbf{x}_t} \left[ \frac{1}{2}\|\mathbf{x}_t\|_1 - \frac{1}{18}\|B\mathbf{x}_t\|_2^2 \right]$$

$$\leq \log 2 - \frac{1}{2} + \mathbf{E}_{\mathbf{x}_t} \left[ \frac{1}{2}\|\mathbf{x}_t\|_1 - \frac{1}{18(12k^2)^2}\|\mathbf{x}_t\|_1^2 \right] \leq \log 2 - \frac{1}{2592k^4}.$$

This inequality means that the stochastic process $\{X_t\}_{t=1}^T$ defined by $X_t = \sum_{j=1}^t \log(1 + \mathbf{r}_t \mathbf{x}_t) - t(\log 2 - \frac{1}{2592k^4})$ is a sub-martingale. From the definition, $\{X_t\}_{t=1}^T$ satisfies $|X_t - X_{t+1}| < \log 3$ for all $t$. Hence, from the Azuma-Hoeffding inequality [2], $X_T$ is bounded as $X_T < 4\sqrt{T}$ with a probability of at least $2/3$. Consequently, we have

$$\sum_{t=1}^T \log(1 + \mathbf{r}_t^\top \mathbf{x}_t) < T(\log 2 - \frac{1}{2592k^4}) + 4\sqrt{T} \leq T\left(\log 2 - \frac{1}{5184k^4}\right),$$

where the last inequality comes from $T \geq (4 \cdot 5184k^4)^2$. This means that the decision is correct with a probability of at least $2/3$. □

## F   Proof of Theorem 6

Let us first consider the following lemma.

**Lemma 8.** *If* $0 \leq p_1 \leq p_2 \leq 1$ *and random variables* $X_1, X_2$ *follow the binomial random distributions* $B(k, p_1), B(k, p_2)$, *respectively, then we have*

$$\mathop{\mathbf{E}}_{X_2 \sim B(k,p_2)}\left[\log\left(1 + \frac{1}{k}X_2\right)\right] - \mathop{\mathbf{E}}_{X_1 \sim B(k,p_1)}\left[\log\left(1 + \frac{1}{k}X_1\right)\right] \geq \frac{p_2 - p_1}{2} \qquad (16)$$

*Proof.* Define $Y_1 = k - X_1$ and $Y_2 = k - X_2$. Then we have $Y_1 \sim B(k, 1 - p_1)$ and $Y_2 \sim B(k, 1 - p_2)$. From the Maclaurin series of $\log(2 - x) = \log 2 - \frac{1}{2}x - \frac{1}{2\cdot2^2}x^2 - \cdots = \log 2 - \sum_{n=1}^\infty \frac{x^n}{n2^n}$, we have

$$\mathop{\mathbf{E}}_{X_2 \sim B(k,p_2)}\left[\log\left(1 + \frac{1}{k}X_2\right)\right] - \mathop{\mathbf{E}}_{X_1 \sim B(k,p_1)}\left[\log\left(1 + \frac{1}{k}X_1\right)\right]$$

$$= \mathop{\mathbf{E}}_{Y_2 \sim B(k,1-p_2)}\left[\log\left(2 - \frac{1}{k}Y_2\right)\right] - \mathop{\mathbf{E}}_{Y_1 \sim B(k,1-p_1)}\left[\log\left(2 - \frac{1}{k}Y_1\right)\right]$$

$$= \sum_{n=1}^\infty \frac{1}{n(2k)^n}\left(\mathop{\mathbf{E}}_{Y_1 \sim B(k,1-p_1)}[Y_1^n] - \mathop{\mathbf{E}}_{Y_2 \sim B(k,1-p_2)}[Y_2^n]\right)$$

$$\geq \frac{1}{2k}\left(\mathop{\mathbf{E}}_{Y_1 \sim B(k,1-p_1)}[Y_1] - \mathop{\mathbf{E}}_{Y_2 \sim B(k,1-p_2)}[Y_2]\right) = \frac{p_2 - p_1}{2}.$$

□

We are now ready to prove Theorem 6.

*Proof of Theorem 6.* We construct an input sequence $\{\mathbf{r}_t\}_{t=1,2,\ldots}$ so that entries $r_{ti}$ follow a uniform random distribution over $\{0, 1\}$ independently. We can show that

$$\mathop{\mathbf{E}}_{\mathbf{r},\mathbf{x}}[R_T(\mathbf{r})] = \mathop{\mathbf{E}}_{\mathbf{r},\mathbf{x}}\left[\max_{S \in \mathcal{S}_k, \mathbf{x} \in \Delta^S} \sum_{t=1}^T \log(1 + \mathbf{r}_t^\top \mathbf{x}) - \sum_{t=1}^T \log(1 + \mathbf{r}_t^\top \mathbf{x}_t)\right] = \Omega\left(\sqrt{T \log \frac{d}{k}}\right) \quad (17)$$

for all algorithms, by means of considering the following two inequalities:

$$\mathop{\mathbf{E}}_{\mathbf{r}_t,\mathbf{x}_t}[\log(1 + \mathbf{r}_t^\top \mathbf{x}_t)] \leq \mathop{\mathbf{E}}_{X_1}\left[\log\left(1 + \frac{1}{k}X_1\right)\right], \qquad (18)$$

$$\mathop{\mathbf{E}}_{\mathbf{r}_t,\mathbf{x}_t}\left[\max_{S \in \mathcal{S}_k, \mathbf{x} \in \Delta^S} \sum_{t=1}^T \log(1 + \mathbf{r}_t^\top \mathbf{x})\right] \geq T \cdot \mathop{\mathbf{E}}_{X_1}\left[\log\left(1 + \frac{1}{k}X_1\right)\right] + \Omega\left(\sqrt{T \log \frac{d}{k}}\right), \qquad (19)$$

where $X_1$ is a binomial random variable following $B(k, 1/2)$.

First, let us prove the inequality (18). Consider a function $\mathbf{x} \mapsto \mathop{\mathbf{E}}_{\mathbf{r}_t}[\log(1 + \mathbf{r}_t^\top \mathbf{x})]$, and suppose $S \in \mathcal{S}_k$. We can then confirm that this is a concave function and that, for the optimization problem

$\arg\max\limits_{\mathbf{x}\in\Delta^S}\mathbf{E}\limits_{\mathbf{r}_t}[\log(1+\mathbf{r}_t^\top\mathbf{x})]$, the vector $\frac{1}{k}\mathbf{1}_S$ is the unique point satisfying KKT conditions, where $\mathbf{1}_S$ stands for the indicator vector of $S$, i.e., $\mathbf{1}_S = [\chi_1,\ldots,\chi_d]^\top$ where $\chi_i = 1$ if $i \in S$ and $\chi_i = 0$ if $i \in [d]\setminus S$. Consequently, we have $\max_{\mathbf{x}\in\Delta^S}\log(1+\mathbf{r}_t^\top\mathbf{x}) = \mathbf{E}\limits_{\mathbf{r}_t}[\log(1+\frac{1}{k}\mathbf{1}_S^\top\mathbf{r}_t)] = \mathbf{E}\limits_{X_1}[\log(1+\frac{1}{k}X_1)]$ since $\mathbf{1}_S^\top\mathbf{r}_t$ follows the binomial distribution $B(k,1/2)$. Since $\mathbf{x}_t \in \Delta^{S_t}$ for some $S_t \in \mathcal{S}_k$ and $S_t, \mathbf{x}_t$ are stochastically independent of $\mathbf{r}_t$, we obtain $\mathbf{E}\limits_{\mathbf{r}_t,\mathbf{x}_t}[\log(1+\mathbf{r}_t^\top\mathbf{x}_t)] \leq \mathbf{E}\limits_{\mathbf{r}_t}[\log(1+\frac{1}{k}\mathbf{r}_t^\top\mathbf{1}_{S_t})] = \mathbf{E}\limits_{X_1}[\log(1+\frac{1}{k}X_1)]$.

Next, let us prove the inequality (19). For each $i \in [d]$, define $r_i := \sum_{t=1}^T r_{ti}$. Since $r_{ti}$ follows a Bernoulli distribution with parameter $1/2$ independently, $r_i$ follows the binomial distribution $B(T,1/2)$. Let $\sigma : [d-k] \to [d-k]$ be a permutation such that $r_{\sigma(1)} \geq r_{\sigma(2)} \geq \cdots \geq r_{\sigma(d-k)}$. Since the posterior random distribution of $r_{ti}$ given $r_i$ is the Bernoulli distribution of parameter $r_i/T$, for $\mathbf{x}_2 = \frac{1}{k}\mathbf{1}_{\{\sigma(1),\sigma(2),\ldots,\sigma(k)\}}$ and for arbitrary constant $s \geq T/2$, we have

$$\mathbf{E}\limits_{\mathbf{r}}\left[\chi_{\{r_{\sigma(k)}\geq s\}} \cdot \sum_{t=1}^T \log(1+\mathbf{r}_t^\top\mathbf{x}_2)\right] \geq \mathbf{E}\limits_{X_2\sim B(k,\frac{s}{T})}\left[\chi_{\{r_{\sigma(k)}\geq s\}} \cdot \sum_{t=1}^T \log\left(1+\frac{1}{k}X_2\right)\right]$$

$$= T \cdot \mathrm{Prob}[r_{\sigma(k)}\geq s] \cdot \mathbf{E}\limits_{X_2\sim B(k,\frac{s}{T})}\left[\log\left(1+\frac{1}{k}X_2\right)\right],$$

where $\chi_A$ stands for the indicator function for arbitrary events $A$. Moreover, since $r_{d-k+1},\ldots,r_d$ are independent of $r_{\sigma(k)}$, for $\mathbf{x}_1 = \frac{1}{k}\mathbf{1}_{\{d-k+1,\ldots,d\}}$, we have

$$\mathbf{E}\limits_{\mathbf{r}}\left[\chi_{\{r_{\sigma(k)}<s\}}\sum_{t=1}^T \log(1+\mathbf{r}_t^\top\mathbf{x}_1)\right] = T \cdot \mathrm{Prob}[r_{\sigma(k)}< s] \cdot \mathbf{E}\limits_{X_1\sim B(k,\frac{1}{2})}\left[\log\left(1+\frac{1}{k}X_1\right)\right].$$

Hence, we obtain

$$\mathbf{E}\limits_{\mathbf{r}}\left[\max_{S\in\mathcal{S}_k,\mathbf{x}\in\Delta^S}\sum_{t=1}^T \log(1+\mathbf{r}_t^\top\mathbf{x})\right]$$

$$= \mathbf{E}\limits_{\mathbf{r}}\left[\chi_{\{r_{\sigma(k)}\geq s\}}\max_{S\in\mathcal{S}_k,\mathbf{x}\in\Delta^S}\sum_{t=1}^T \log(1+\mathbf{r}_t^\top\mathbf{x})\right] + \mathbf{E}\limits_{\mathbf{r}}\left[\chi_{\{r_{\sigma(k)}<s\}}\max_{S\in\mathcal{S}_k,\mathbf{x}\in\Delta^S}\sum_{t=1}^T \log(1+\mathbf{r}_t^\top\mathbf{x})\right]$$

$$\geq \mathbf{E}\limits_{\mathbf{r}}\left[\chi_{\{r_{\sigma(k)}\geq s\}}\sum_{t=1}^T \log(1+\mathbf{r}_t^\top\mathbf{x}_2)\right] + \mathbf{E}\limits_{\mathbf{r}}\left[\chi_{\{r_{\sigma(k)}<s\}}\sum_{t=1}^T \log(1+\mathbf{r}_t^\top\mathbf{x}_1)\right]$$

$$\geq T \cdot \mathrm{Prob}[r_{\sigma(k)}\geq s] \cdot \mathbf{E}\limits_{X_2}\left[\log\left(1+\frac{1}{k}X_2\right)\right] + T \cdot \mathrm{Prob}[r_{\sigma(k)}\leq s] \cdot \mathbf{E}\limits_{X_1}\left[\sum_{t=1}^T \log\left(1+\frac{1}{k}X_1\right)\right]$$

$$\geq T \cdot \mathbf{E}\limits_{X_1}\left[\log\left(1+\frac{1}{k}X_1\right)\right] + T \cdot \mathrm{Prob}[r_{\sigma(k)}\geq s] \cdot \frac{1}{2}\cdot\left(\frac{s}{T}-\frac{1}{2}\right),$$

where $X_1 \sim B(k,\frac{1}{2})$, $X_2 \sim B(k,\frac{s}{T})$ and the last inequality comes from Lemma 8. We now can show that we have $\mathrm{Prob}[r_{\sigma(k)}\geq s] = \Omega(1)$ for $s = \frac{T}{2}+\Omega(\sqrt{T\log\frac{d}{k}})$, which proves (19). Let $F : \mathbb{R} \to [0,1]$ denote the cumulative distribution function of $B(T,1/2)$, i.e., $F(x) = \mathrm{Prob}[r_i \leq x]$. From a standard concentration lemma of a binomial distribution (see, e.g., Proposition 7.3.2 in [25]), we have $F(\frac{T}{2}+t) \leq 1 - \frac{1}{15}\exp\left(-16\frac{t^2}{T}\right)$. Hence, setting $t = \frac{1}{4}\sqrt{T\log\frac{d-k}{15k}}$, we obtain

$$\mathrm{Prob}\left[r_{\sigma(k)}\geq\frac{T}{2}+t\right] = \mathrm{Prob}\left[F(r_{\sigma(k)})\geq F\left(\frac{T}{2}+t\right)\right]$$

$$\geq \mathrm{Prob}\left[F(r_{\sigma(k)})\geq 1 - \frac{1}{15}\exp\left(-16\frac{t^2}{T}\right)\right] = \mathrm{Prob}\left[F(r_{\sigma(k)})\geq 1 - \frac{k}{d-k}\right]$$

Since $F(r_i)$ follows the uniform distribution on $[0,1]$ independently, $F(r_{\sigma(k)})$ follows the probability distribution of the order statistic sampled from the standard uniform distribution, which is the beta

distribution $\text{Beta}(d - 2k + 1, k)$ (see, e.g., [15]). This means that $\text{Prob}[1 - F(r_{\sigma(k)}) \leq \frac{k}{d-k}] \geq 1/2$. Combining the above two inequalities, for $s = \frac{T}{2} + \frac{1}{4}\sqrt{T \log \frac{d-k}{15k}}$, we have

$$
\mathbf{E}_{\mathbf{r}} \left[ \max_{S \in \mathcal{S}_k, \mathbf{x} \in \Delta^S} \sum_{t=1}^T \log(1 + \mathbf{r}_t^\top \mathbf{x}) \right]
$$

$$
\geq T \cdot \mathbf{E}_{X_1} \left[ \log\left(1 + \frac{1}{k}X_1\right) \right] + \frac{1}{2}\text{Prob}[r_{\sigma(k)} \geq s] \cdot \left( s - \frac{T}{2} \right)
$$

$$
\geq T \cdot \mathbf{E}_{X_1} \left[ \log\left(1 + \frac{1}{k}X_1\right) \right] + \frac{1}{16}\left( \sqrt{T \log \frac{d-k}{15k}} \right)
$$

$$
\geq T \cdot \mathbf{E}_{X_1} \left[ \log\left(1 + \frac{1}{k}X_1\right) \right] + \Omega\left( \sqrt{T \log \frac{d}{k}} \right),
$$

where the last inequality comes from $d \geq 17k$. Consequently, we obtain (19). From (18) and (19) we have (17). □

## G  Proof of Theorem 7

**Lemma 9.** *For arbitrary $\epsilon \in [0, 1/2]$, let $z_0, z_1, \ldots, z_k \in \{-1, 1\}$ be independent random variables such that $z_0 = \begin{cases} 1 & \text{w.p. } 1/2 - \epsilon \\ -1 & \text{w.p. } 1/2 + \epsilon \end{cases}$ and $z_i = \begin{cases} 1 & \text{w.p. } 1/2 \\ -1 & \text{w.p. } 1/2 \end{cases}$ for $i = 1, \ldots, d$. Set $X_1 = \sum_{i=1}^k z_i$ and $X_2 = \sum_{i=1}^{k-1} z_i + z_0 \prod_{j=1}^{k-1} z_j$. We then have*

$$
\mathbf{E}\left[ \log\left(2 - \frac{1}{k}X_2\right) \right] - \mathbf{E}\left[ \log\left(2 - \frac{1}{k}X_1\right) \right] \geq \frac{2\epsilon}{k(2k)^k}. \tag{20}
$$

*Proof.* Denote $w = z_0 \prod_{j=1}^{k-1} z_j$. Let $n_1, n_2, \ldots, n_k$ be arbitrary non-negative integers. Set $m_i$ to be $n_i$ modulo 2, i.e., $m_i = 0$ if $n_i$ is even and $m_i = 1$ if $n_i$ is odd, for $i = 1, \ldots, k$. We then have

$$
\mathbf{E}[z_1^{n_1} z_2^{n_2} \cdots z_{k-1}^{n_{k-1}} z_k^{n_k}] = \mathbf{E}[z_1^{m_1} z_2^{m_2} \cdots z_{k-1}^{m_{k-1}} z_k^{m_k}] = \begin{cases} 1 & \text{if } m_1 = \cdots = m_k = 0 \\ 0 & \text{otherwise} \end{cases}
$$

$$
\mathbf{E}[z_1^{n_1} z_2^{n_2} \cdots z_{k-1}^{n_{k-1}} w^{n_k}] = \mathbf{E}[z_1^{m_1} z_2^{m_2} \cdots z_{k-1}^{m_{k-1}} w^{m_k}] = \begin{cases} 1 & \text{if } m_1 = \cdots = m_k = 0 \\ -2\epsilon & \text{if } m_1 = \cdots = m_k = 1 \\ 0 & \text{otherwise} \end{cases},
$$

which means $\mathbf{E}[z_1^{n_1} \cdots z_{k-1}^{n_{k-1}} z_k^{n_k}] \geq \mathbf{E}[z_1^{n_1} \cdots z_{k-1}^{n_{k-1}} w^{n_k}]$. Hence, $X_1 = \sum_{i=1}^k z_i$ and $X_2 = \sum_{i=1}^{k-1} z_i + w$ satisfies $\mathbf{E}[X_1^n] \geq \mathbf{E}[X_2^n]$ for all non-negative integers $n$ and $\mathbf{E}[X_1^k] - \mathbf{E}[X_2^k] = 2\epsilon$. From the Maclaurin series of $\log(2 - x) = \log 2 - \sum_{n=1}^\infty \frac{x^n}{n2^n}$, we have

$$
\mathbf{E}\left[ \log\left(2 - \frac{1}{k}X_2\right) \right] - \mathbf{E}\left[ \log\left(2 - \frac{1}{k}X_1\right) \right] = \sum_{n=1}^\infty \frac{1}{n(2k)^n}(\mathbf{E}[X_1^n] - \mathbf{E}[X_2^n])
$$

$$
\geq \frac{1}{k(2k)^k}(\mathbf{E}[X_1^k] - \mathbf{E}[X_2^k]) = \frac{2\epsilon}{k(2k)^k}.
$$

□

**Proof of Theorem 7**

*Proof.* For each $S^* \in \mathcal{S}_k$, we define a random distribution $D_{S^*}$ on $\{-1, 1\}^d$ so that $\mathbf{z} = [z_1, \ldots, z_d]^\top \sim D_{S^*}$ satisfies

$$
\prod_{i \in S^*} z_i = \begin{cases} 1 & \text{w. p. } 1/2 - \epsilon \\ -1 & \text{w. p. } 1/2 + \epsilon \end{cases}, \qquad \prod_{i \in S} z_i = \begin{cases} 1 & \text{w. p. } 1/2 \\ -1 & \text{w. p. } 1/2 \end{cases} \quad (S \in 2^{[d]} \setminus \{\emptyset, S^*\}). \tag{21}
$$

Such a distribution can be constructed as follows: fix an index $i^* \in S^*$ and, for $i \in [d] \setminus \{i^*\}$, let $z_i = \begin{cases} 1 & \text{w. p. } 1/2 \\ -1 & \text{w. p. } 1/2 \end{cases}$ and $z_0 = \begin{cases} 1 & \text{w. p. } 1/2 - \epsilon \\ -1 & \text{w. p. } 1/2 + \epsilon \end{cases}$ independently. Define $z_{i^*} = z_0 \prod_{i \in S^* \setminus \{i^*\}} z_i$. Suppose that the input sequence $\mathbf{r}_t$ is given by $\mathbf{r}_t = \mathbf{1} - \mathbf{z}_t$, where $\mathbf{z}_t \sim D_S^*$ independently for $t = 1, 2, \ldots, T$. If $\mathbf{z}$ follows $D_S^*$, for any $S \in \mathcal{S}_k \setminus \{\emptyset S^*\}$, $\mathbf{z}|_S$ follows the uniform distribution on $\{-1, 1\}^S$, and hence, we have $\max_{\mathbf{x} \in \Delta^{S'}} \mathop{\mathbf{E}}_{\mathbf{z}_t \sim D_{S^*}} \left[ \log \left( 1 + \mathbf{r}_t^\top \mathbf{x} \right) \right] = \mathop{\mathbf{E}}_{\mathbf{z}_t \sim D_{S^*}} \left[ \log \left( 1 + \frac{1}{k} \mathbf{r}_t^\top \mathbf{1}_S \right) \right]$. From Lemma 9, for $S \in \mathcal{S}_k \setminus \{\emptyset S^*\}$, we have

$$\mathop{\mathbf{E}}_{\mathbf{z}_t \sim D_{S^*}} \left[ \log \left( 1 + \frac{1}{k} \mathbf{r}_t^\top \mathbf{1}_{S^*} \right) \right] - \max_{\mathbf{x} \in \Delta^S} \mathop{\mathbf{E}}_{\mathbf{z}_t \sim D_{S^*}} \left[ \log \left( 1 + \mathbf{r}_t^\top \mathbf{x} \right) \right]$$

$$\geq \mathop{\mathbf{E}}_{\mathbf{z}_t \sim D_{S^*}} \left[ \log \left( 1 + \frac{1}{k} \mathbf{r}_t^\top \mathbf{1}_{S^*} \right) \right] - \mathop{\mathbf{E}}_{\mathbf{z}_t \sim D_{S^*}} \left[ \log \left( 1 + \frac{1}{k} \mathbf{r}_t^\top \mathbf{1}_S \right) \right] \geq \frac{2\epsilon}{k(2k)^k}. \tag{22}$$

Since any randomized algorithm is equivalent to an a priori random choice from the set of all deterministic strategies, and since the input defined above is oblivious to the output of the algorithm, it suffices to prove a lower bound on the expected regret of any *deterministic* algorithm (this is not crucial for the proof but simplifies the notation). We consider an arbitrary deterministic algorithm and let $\{(S_t, \mathbf{x}_t)\}_{t=1}^T$ denote the output for the random input sequence $\{\mathbf{r}_t\}_{t=1}^T$ given by $\mathbf{r}_t = \mathbf{1} + \mathbf{z}_t$ and $\mathbf{z}_t \sim D_{S^*}$. Let $N_S$ be a random variable denoting the number of $t \in [T]$ such that $S_t = S$, i.e., $N_S = |\{t \in [T] \mid S_t = S\}|$. From the equation (22), we have

$$\mathop{\mathbf{E}}_{\mathbf{z}_t \sim D_{S^*}} [R_T] \geq \sum_{t=1}^T \mathop{\mathbf{E}}_{\mathbf{z}_t \sim D_{S^*}} \left[ \log \left( 1 + \frac{1}{k} \mathbf{r}_t^\top \mathbf{1}_S \right) \right] - \sum_{t=1}^T \mathbf{E} \left[ \max_{S_t} \mathop{\mathbf{E}}_{\mathbf{x} \in \Delta^{S_t}} \mathop{\mathbf{E}}_{\mathbf{z}_t \sim D_{S^*}} \left[ \log \left( 1 + \mathbf{r}_t^\top \mathbf{x} \right) \right] \right]$$

$$\geq \left( T - \mathop{\mathbf{E}}_{\mathbf{z}_t \sim D_{S^*}} [N_{S^*}] \right) \frac{2\epsilon}{k(2k)^k}. \tag{23}$$

Let us evaluate $\mathop{\mathbf{E}}_{\mathbf{z}_t \sim D_{S^*}} [N_{S^*}]$. Define $D_0$ to be the uniform probabilistic distribution on $\{-1, 1\}^d$. Then, for all $S \in \mathcal{S}_k \setminus \{\emptyset, S^*\}$, we have $D_{S^*}|_S = D_0|_S$, i.e., if $\mathbf{z} \sim D_{S^*}$ and $\mathbf{z}' \sim D_0$, then $\mathbf{z}|_S$ and $\mathbf{z}'|_S$ follows the same distribution (a uniform distribution on $\{-1, 1\}^S$). Hence, in the same way as in Lemma A.1. of [5], we can show that

$$\mathop{\mathbf{E}}_{\mathbf{z}_t \sim D_{S^*}} [N_{S^*}] - \mathop{\mathbf{E}}_{\mathbf{z}_t \sim D_0} [N_{S^*}] \leq \frac{T}{\sqrt{2}} \sqrt{\mathop{\mathbf{E}}_{\mathbf{z}_t \sim D_0} [N_{S^*}] \cdot \mathrm{KL}\left( D_0|_{S^*} \,\middle\|\, D_{S^*}|_{S^*} \right)} \tag{24}$$

where $\mathrm{KL}(P \| Q) = \mathop{\mathbf{E}}_P (\log \frac{\mathrm{d}P}{\mathrm{d}Q})$ is the Kullback-Leibler divergence. The chain rule for relative entropy (see, e.g., Theorem 2.5.3 of [11]) gives, for $S^* = \{i_1, \ldots i_k\}$,

$$\mathrm{KL}\left( D_0|_{S^*} \,\middle\|\, D_{S^*}|_{S^*} \right) = \mathrm{KL}(\mathrm{Prob}_{\mathbf{z} \sim D_0}[(z_i)_{i \in S^*}] \,\|\, \mathrm{Prob}_{\mathbf{z} \sim D_{S^*}}[(z_i)_{i \in S^*}])$$

$$= \sum_{j=1}^k \mathop{\mathbf{E}}_{(z_{i_s})_{s<j}} \left[ \mathrm{KL}(\mathrm{Prob}_{\mathbf{z} \sim D_0}[z_{i_j} \mid (z_{i_s})_{s<j}] \,\|\, \mathrm{Prob}_{\mathbf{z} \sim D_{S^*}}[z_{i_j} \mid (z_{i_s})_{s<j}]) \right]$$

$$= \mathop{\mathbf{E}}_{(z_{i_s})_{s<k}} \left[ \mathrm{KL}(\mathrm{Prob}_{\mathbf{z} \sim D_0}[z_{i_k} \mid (z_{i_s})_{s<k}] \,\|\, \mathrm{Prob}_{\mathbf{z} \sim D_{S^*}}[z_{i_k} \mid (z_{i_s})_{s<k}]) \right]$$

$$= -\frac{1}{2} \log(1 - 4\epsilon^2). \tag{25}$$

In the above equations, the third equality holds because $\mathrm{Prob}_{\mathbf{z} \sim D_0}[z_{i_j} \mid (z_{i_s})_{s<j}]$ and $\mathrm{Prob}_{\mathbf{z} \sim D_{S^*}}[z_{i_j} \mid (z_{i_s})_{s<j}]$ are equal to the Bernoulli distribution of parameter $1/2$ for $j < k$. The last equality holds because $\mathrm{Prob}_{\mathbf{z} \sim D_0}[z_{i_k} \mid (z_{i_s})_{s<k}]$ follows Bernoulli distribution of parameter $1/2$ and $\mathrm{Prob}_{\mathbf{z} \sim D_{S^*}}[z_{i_k} \mid (z_{i_s})_{s<k}]$ follows Bernoulli distribution of parameter $1/2 + \epsilon$ or $1/2 - \epsilon$. Combining (23), (24) and (25), we have

$$\mathop{\mathbf{E}}_{\mathbf{z}_t \sim D_{S^*}} [R_T] \geq \left( T - \mathop{\mathbf{E}}_{\mathbf{z}_t \sim D_0} [N_{S^*}] - \frac{T}{2} \sqrt{- \mathop{\mathbf{E}}_{\mathbf{z}_t \sim D_0} [N_{S^*}] \log(1 - \epsilon^2)} \right) \frac{2\epsilon}{k(2k)^k}.$$

Suppose that $S^*$ is chosen at random uniformly from $\mathcal{S}_k$, before play begins. Then, from the above inequality, the expected regret is bounded as

$$\mathop{\mathbf{E}}_{S^*, \mathbf{z}_t \sim D_{S^*}}[R_T] \geq \frac{1}{|\mathcal{S}_k|} \sum_{S^* \in \mathcal{S}_k} \left( T - \mathop{\mathbf{E}}_{\mathbf{z}_t \sim D_0}[N_{S^*}] - \frac{T}{2} \sqrt{- \mathop{\mathbf{E}}_{\mathbf{z}_t \sim D_0}[N_{S^*}] \log(1 - 4\epsilon^2)} \right) \frac{2\epsilon}{k(2k)^k}$$

$$\geq \left( T - \frac{T}{|\mathcal{S}_k|} - \frac{T}{2} \sqrt{-\frac{T}{|\mathcal{S}_k|} \log(1 - 4\epsilon^2)} \right) \frac{2\epsilon}{k(2k)^k},$$

where the second inequality comes from $\sum_{S \in \mathcal{S}_k} \mathop{\mathbf{E}}_{\mathbf{z}_t \sim D_0}[N_S] = T$ and $\sum_{S \in \mathcal{S}_k} \sqrt{\mathop{\mathbf{E}}_{\mathbf{z}_t \sim D_0}[N_S]} \leq \sqrt{T|\mathcal{S}_k|}$. Using the inequality $-\log(1 - x) \leq x/2$ for $x \in [0, 1/4]$, we have

$$\mathop{\mathbf{E}}_{S^*, \mathbf{z}_t \sim D_{S^*}}[R_T] \geq T \left( 1 - \frac{1}{|\mathcal{S}_k|} - \epsilon \sqrt{\frac{T}{2|\mathcal{S}_k|}} \right) \frac{2\epsilon}{k(2k)^k}$$

for $\epsilon \in [0, 1/4]$. By setting $\epsilon = \min\{1/4, \frac{1}{2}\sqrt{\frac{|\mathcal{S}_k|}{T}}\}$, we obtain

$$\mathop{\mathbf{E}}_{S^*, \mathbf{z}_t \sim D_{S^*}}[R_T] = \Omega \left( \min \left\{ \frac{T}{k(2k)^k}, \sqrt{\frac{T|\mathcal{S}_k|}{k^2(2k)^{2k}}} \right\} \right) \geq \Omega \left( \min \left\{ \frac{T}{k(2k)^k}, \sqrt{T \left( \frac{d}{5k^3} \right)^k} \right\} \right),$$

(26)

where the second inequality follows from $|\mathcal{S}_k| = \binom{d}{k} \geq (\frac{d}{k})^k$ and $k^2 = O((\frac{5}{4})^k)$.

Consider an arbitrary randomized algorithm and let $\lambda$ denote the algorithm's internal randomization. Then, since $\lambda$ is probabilistically independent from $S^*, \mathbf{r}$ and (26) for all deterministic algorithms, we have

$$\mathop{\mathbf{E}}_{S^*, \{\mathbf{r}\}} \left[ \mathop{\mathbf{E}}_{\lambda}[R_T] \right] = \mathop{\mathbf{E}}_{\lambda} \left[ \mathop{\mathbf{E}}_{S^*, \{\mathbf{r}\}}[R_T] \right] = \Omega \left( \min \left\{ \frac{T}{k(2k)^k}, \sqrt{T \left( \frac{d}{5k^3} \right)^k} \right\} \right).$$

$\square$