[Reviews · NeurIPS 2018]

Reviewer 1



This paper investigate online portfolio under cardinality constraints. Matching upper/lower bounds in full-information/bandit settings are provided. The paper is well-written. I could not check the details in the paper because it is too technical for me given the limited review time. But viewing the results, the authors have almost clarify the landscape of this problem in both regret upper/lower bounds and computational complexity. Given that there are relatively less literature about computational lower bounds, I would consider the technical contributions in this paper significant.

Reviewer 2



Summary The paper studies the online portfolio selection problem under cardinality constraints and provides two algorithms that achieve sublinear regret. One algorithm handles the full information setting and the other algorithm handles the bandit feedback setting. Furthermore, the paper provides lower bounds for both the full information and bandit feedback settings. The approach that both algorithms take is to split the problem into two learning problems. One problem is to learn the optimal combination of assets and the other problem is to learn the optimal portfolio. To learn the optimal combination of assets a version of either the multiplicative weights algorithm (full information) or exp3 (bandit feedback) is used. To learn the optimal portfolio the Follow The Approximation Leader (FTAL) algorithm is used. Finally, the papers provides three experiments in which the two new algorithms and three baseline algorithms are evaluated on one synthetic data set and two real world data sets. Strengths and weaknesses The problem that is solved in this paper has nice practical applications. The analysis of the algorithm was elegant and surprisingly simple after splitting the problem into two parts. After the splitting in two parts the analysis almost directly follows from applying standard algorithms with their standard regret bounds. On the downside, since for each combination of assets a version of FTAL is run the proposed algorithm for the full information has exponential runtime in the dimension. However, the authors show that the runtime cannot be improved while preserving the regret bound. Since the FTAL is run locally in the sense that each combination of assets gets its own instance of FTAL it would be interesting to see if the dependency on the constants C1, C2, C3, C4, and C5 in the regret bound can also depend on the optimal combination of assets. I do not understand the reason for choosing these particular baseline algorithms. It seems very restrictive to only consider x_{t,i} \in {0, 1}, as FTAL can consider x_{t,i} \in [0, 1]. Therefore, the conclusion that algorithms 1 and 2 outperform the baseline algorithms is not very surprising. It appears a more informative baseline algorithm would be to use MWU/exp3 with gradient descent or exponential weights with a continuous prior. The presentation of the material was reasonably clear, although the section on computational complexity is quite dense. Minor comments - Figures 1-3: the colors of MWU_disc and Algorithm 2 are difficult to discern, perhaps use black instead of purple. - the statement “our algorithms are superior to others” seems a bit strong. - line 349: the low of large number -> the law of large numbers. The authors response cleared up some of the confusion regarding the simulations.

Reviewer 3



This paper mainly studied online portfolio selection with a cardinality constraint. In full information and partial information cases, authors proposed corresponding algorithms based on existing MWU, FTAL and EXP3 algorithms, which were then proved to be nearly optimal through their regret performances and respective lower bounds. In spite of the exponential running time of proposed algorithms, authors also proved a computational lower bound, which showed there was no polynomial time algorithm could achieve sublinear regret. Further experiments exhibited the efficacy of proposed algorithms. In general, this paper is well-written, and the problem itself looks interesting. Proposed algorithms are natural to be thought of in the first place, which are combinations of existing best algorithms in respective sub-problems. The main highlight of paper is almost completely solving this problem, i.e. proving corresponding lower bounds as well as a computational lower bound, which shows the near optimality of proposed algorithms, though it doesn’t look so surprising in some sense. Given above clear results, I still feel kind of confused about some places: 1. In line 134, FTAL used in this paper is a little different from the original FTAL, which doesn’t have an additional square regularizer, so what is the use of this regularizer and is it necessary? 2. In corollary 5, it seems there should be some constraints about the range of k, otherwise for k = d, ONS step can achieve sublinear regret in polynomial time. Please make it clear. Other minor comments: 1. In the upper bound of combination constraint in table 1, it may be better to use equation (5), as we can see O(d \log T) regret from equation (5) when k = d (i.e. |S| = 1); 2. In line 71, d=\Omega(k) is kind of misleading, as for d =k, we can obtain O(d \log T) regret, which disagrees with the lower bound \sqrt{T}; 3. In line 88 and 95, the regret should be O(d \log T); 4. In line 428, positions of Y_1 and Y_2 are reversed in the last two lines. Update: Thanks for authors' response, and it solved my confusion.